# The Human Body as an Ethanol-Producing Bioreactor—The Forensic Impacts

## Ivan Šoša

Department of Anatomy, Faculty of Medicine, University of Rijeka, 51000 Rijeka, Croatia; ivan.sosa@uniri.hr

**Abstract:** Auto-brewery syndrome (ABS), also called gut fermentation syndrome, is an extremely infrequent but also underrecognized disorder where ethanol is produced endogenously, similar to a typical bioreactor. The reliability of forensic alcohol analysis results is frequently challenged as the ethanol concentration in the breath, blood, and/or urine constitutes important evidence for prosecuting drivers under the influence of the alcohol. This further emphasizes the need to understand ABS, as in legal proceedings it is often presented as grounds for acquittal due to the concept that the findings could have corresponded to endogenously produced ethanol. However, this rare and underdiagnosed medical condition should not be considered as purely a lawyer's favorite argument. Manifestations of ABS can have a severe impact on a patient's life and pose social consequences as well. Unfortunately, barely anything has been unearthed, and aspects such as genetic susceptibility, gut-mucus-eating microorganisms, and fecal microbiome transplantation were reviewed for the first time in this context. The framework of this review was not limited to the gut microbiota exclusively; moreover, the overgrowth of microorganisms is linked to the use of antibiotics. Studies have indicated that carbohydrate fermentation occurs in locations other than in intra-intestinal flora. Accordingly, the literature was searched for cases of patients with ABS with yeast infections in their genitourinary or oral systems.

**Keywords:** alcohol; auto-brewery syndrome; bioreactor; fermentation; microbiome





## 1. Introduction

Bioreactors are devices or systems maintaining a biologically active environment in which a chemical process that involves organisms or biochemically active substances is carried out [1]. Even though there is enough evidence to support the concept of auto-brewery syndrome (ABS; sometimes referred to as gut fermentation syndrome, endogenous ethanol fermentation, or drunkenness disease), parties in the legal process do not employ this strategy frequently [2–5]. In the interplay of the gut microbiota and endogenous ethanol production, the production of ethanol through the endogenous fermentation of carbohydrates is an inevitable result [6–9]. Even though ABS would seem to be a rarely diagnosed condition, legal experts should be aware that it exists and that it could require a different legal treatment. Greater attention should be paid to this entity after a report of the production of alcohol in a rather atypical part of the digestive system, i.e., the oral cavity [10].

Unexplained intoxication symptoms—such as disorientation, dizziness, and ataxia—are common presentations of ABS (Table 1). Clearly, alcohol consumption must be excluded.

**Table 1.** Symptoms of auto-brewery syndrome [accoeding to: Paramsothy et al., 2023] [11].

| Systems | Symptoms |
| --- | --- |
| General | Unexplained intoxication, Glassy eyes, Smell of alcohol in breath, Chronic fatigue. |
| Nervous System | Memory loss, Mental status changes, Recurrent seizures, Slurred speech, Incoherent speech, Difficulty in articulation, Blurred vision, Dizziness, Disorientation, Ataxia |
| Gastrointestinal System | Bloating, Belching, Nausea, Vomiting. |
| Musculoskeletal System | Poor coordination, frequent falls, Stumbling gait. |

"Alcohol-producing" processes may be succinctly abstracted to ethanol's constant formation from acetaldehyde in the human body through various metabolic processes [12]. In such cases where it is not introduced from the environment, it is called "endogenous ethanol" [9,13–16].

Alcohol metabolism is influenced by the host's state of energy, nutrition, and hormones, but it essentially basically involves three simple steps. In the first step, ethanol is oxidized to the product acetaldehyde. Afterward, acetaldehyde is oxidized to acetate; generally speaking, much of the acetaldehyde produced from the oxidation of alcohol is oxidized in the liver to acetate (circulating levels of acetaldehyde are low under normal conditions). In peripheral tissues, this is activated by a key Acetyl CoA. Two key enzymes are included in these steps (Figure 1), namely, alcohol dehydrogenase (ADH) and aldehyde dehydrogenase (ALDH), which, in healthy individuals, are involved in the breakdown of ethanol and acetaldehyde into harmless acetate. Acetyl CoA is also the key metabolite produced from all major nutrients, i.e., carbohydrates, fat, and excess protein [17].

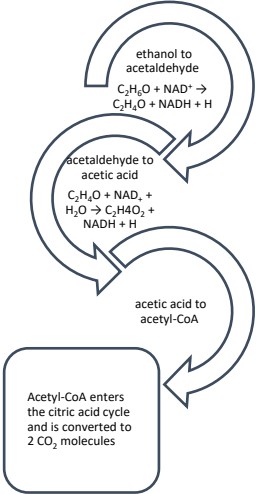

**Figure 1.** Alcohol metabolism in the human body comprises three crucial steps: after alcohol is metabolized by by ADH into a toxic acetaldehyde, it is then oxidized to acetic acid and acetyl-CoA. Acetyl-CoA is free to enter directly into the citric acid cycle. However, under alcoholic conditions, the citric acid cycle is stalled by the oversupply of NADH derived from ethanol oxidation.

Accordingly, the resulting carbon atoms from alcohol are the same products produced from the oxidation of carbohydrates, fat, and protein. Dietary recommendations for adult humans advise that carbohydrates make up 45% to 65% of total daily calories [18]. This quota is easy to accomplish since the molecules are found in a wide array of food products. However, there are carbohydrates available to organisms besides those in food [19].

The incentive behind the current manuscript was to provide a review of cases of endogenous ethanol production and their respective forensic impacts. This has a significant

value in clinical settings, legal proceedings, and forensic science. This review will be an innovative first effort to deal with ABS in the context of the gut–liver–brain axis and mucus-eating microorganisms and deliberate on innovative treatment approaches. This study aims to expand upon the current reviews on the topic.

## 2. A Systematic Review

To conduct this review, the Web of Science Core Collection, PubMed, and Scopus databases were searched for entries with "auto-brewery syndrome", "endogenous ethanol production", or "gut fermentation syndrome" contained in the title. All the resulting entries were included in the flowchart irrespective of whether they were published journal articles, scientific meeting abstracts, entries from a study register, reports of a clinical study, dissertations, unpublished material, government reports, or any other documents providing relevant information. The final literature review considered published journal articles only.

The initial search (with "OR" as an operator) yielded 19 results from PubMed, 58 from the Web of Science Core Collection, and 44 document results from the Scopus database.

From the total number of 121 reports, 9 duplicates were excluded, and 1 record was identified as a non-English contribution. This amounted to a total of 10 records excluded by the automation tool. Another 34 records were manually excluded, as they were either unrelated to the topic or identified as duplicates when scrutinized. A total of 53 journal articles were used for this systematic review, and the Preferred Reporting Items for Systematic Reviews and Meta-Analyses (PRISMA) flow diagram of this search can be observed in Figure 2. PRISMA diagrams can be used in the reporting of reviews evaluating, e.g., etiology, prevalence, diagnosis, or prognosis [20].

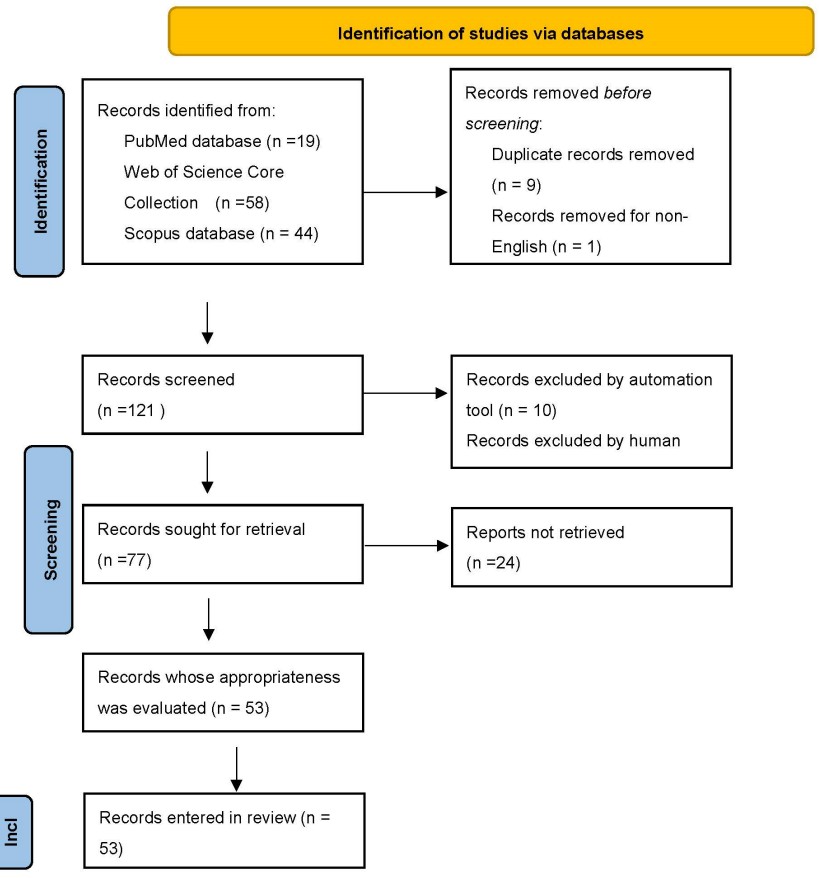

**Figure 2.** PRISMA flow diagram for the systematic review of PubMed, Web of Science, and the Scopus databases; "auto-brewery syndrome", "endogenous ethanol production", or "gut fermentation syndrome" were the potential search terms contained in the titles.

## 2.1. Blood Ethanol Levels

Blood ethanol levels in healthy individuals may vary between 0 and 0.7 mmol/L, and in patients with diabetes or cirrhosis, even higher levels are observed [4,14,21–24]. The first reliable (chromatographic) article published on the concentrations of endogenously produced ethanol was that written by Lester from 1962. He established that the "normal" blood alcohol concentration (BAC) of humans, without exogenous intake, ranges from 0.0 to 0.071 µg/dL [22]. These concentrations result from the permanent production of acetaldehyde. The human body constantly forms ethanol acetaldehyde through various metabolic processes [14]. Later reports have claimed to find body fluids with abnormally high concentrations of ethanol in apparently healthy individuals; however, these reports happen to suffer from methodological deficiencies, a complete lack of appropriate control trials, or the use of overly comprehensive methods of analysis. The majority of the searched literature consisted of case reports or case series. Even case reports may be used to develop or write a systematic review. A special protocol for systematic reviews that use reports or studies of cases/case series was consulted [25]. This protocol is especially useful in disciplines like forensic medicine, where case reports/studies are a common practice.

For instance, besides Lester's finding of endogenous ethanol production [22], more recent researchers, such as Al-Awadhi et al., provided some new perspectives on ABS and established that endogenous ethanol production could occur at 0.04 mg/dL [4]. Yet another large-sample study from Saudi Arabia yielded blood ethanol concentrations so modest that they were far too low to have any forensic significance [26].

With reliable means, e.g., gas chromatography (GC), it was determined that the concentrations of endogenous ethanol in the peripheral venous blood of healthy individuals, as well as those afflicted by specific metabolic conditions (diabetes, hepatitis, and cirrhosis), hardly surpassed 0.08 mg/dL [27].

What about all those other cases? Forensics and legal professionals often settle cases with any plausible drunk-driving defense strategy [3,5,6,8,28]; however, a consensus stating that this metabolic disturbance is rare should be reached [5,6]. Yeast, certain mold fungi species, and several bacterial strains are capable of producing lactic acid and ethanol when in anaerobic conditions (in particular, Candida albicans and Saccharomyces cerevisiae can anaerobically convert carbohydrates into endogenous ethanol and carbon dioxide) [6]. They employ the Entner–Doudoroff pathway (typically prokaryotic heterofermentation) [29,30]. The fermentation process begins via Embden–Meyerhof's glycolysis, which is typical for eukaryotes [15]. Moreover, resulting from additional reactions involving the Ehrlich pathway, higher alcohols are produced [9,31]. Unfortunately, blood ethanol concentration is considered fundamental evidence in cases of driving under the influence of alcohol, regardless of its origin [5,28,32].

The gut fermentation of carbohydrates in the human body, acting as a bioreactor, in amounts sufficient to produce the effects of intoxication is relatively rare [8,23,33,34], but the gut microbiota is present even in the more subtle form of an endogenous brew [35–37]. In this biological process, which is called alcoholic fermentation, sugars are converted into energy for cells, with ethanol and carbon dioxide produced as by-products [38]. The average person passes about 0.5 L of gas a day, which is a by-product of fermentation; this value is far from enough to produce an intoxicating effect. As a rough estimate, increasing this value from 0.00 to 1.0 g/kg in two hours would require a person to pass approximately 20 L of gas during a given period [4,15,39,40]. Alcoholic beverages and ethanol fuel employ this very process [5,41,42].

Thus, it is reasonable to conclude that the microbiota has a range of hand-in-glove microorganisms that have learned to exist with us [37]. Interestingly, organisms known to ferment sugar and generate copious amounts of gas, particularly bacilli and Gram-negative cocci, were found in samples from an autopsy of a boy from Africa who died after the perforation of the back of the wall of the abdominal cavity due to extreme gas distention [43].

## 2.2. The Argument for the Hypothetical Endogenous Origin of Alcohol

Due to its hydrophilic nature, once ethanol is introduced into the bloodstream, it is evenly dispersed through the volume of water contained therein. The faster a person drinks alcohol, the quicker they become intoxicated [5,9,44]. Moderate consumption of ethanol decreases stress and increases feelings of happiness and well-being. It may also reduce the risk of coronary heart disease [45]. On the other hand, heavy drinking may cause addiction, lead to a myriad of diseases associated with alcoholism, and increase the likelihood of suffering from all types of injury [46]. Patients with ABS typically show many symptoms of ethyl alcohol poisoning while denying its intake. However, they all report the consumption of carbohydrate-rich diets (white bread, pastries, potatoes, rice, and pasta) [33,47].

The unsatisfying component of reading the various case studies involving ABS is the lack of a timeline of whether there is a gradual increase in blood alcohol content [3,5,8,24,48–52]. In people who consume alcohol, the presence of food in the stomach should be considered, as it slows down the absorption of alcohol [44,53]. In addition, it should be clearly stated that naturally produced ethanol is delivered directly into the bloodstream, bypassing both the fermentation and metabolization processes [33].

## 2.3. Non-Alcoholic Food-Derived Ethanol

People have savored fermented foods since antiquity. Foods such as yogurt, sauerkraut, and sourdough bread are all products of fermentation. Nevertheless, the fermented non-alcoholic beverage market constantly flourishes. Although these products are not designed to be alcoholic, their ethanol content can vary [54]. Fermentation is a natural process that turns sugars into ethanol [29].

Even if fermentation is disregarded, an alcohol derived from food could cause a problem in forensic cases. The alcohol content in foods has been analyzed in consideration of the human daily average food consumption and food-derived blood alcohol concentrations in reference to the data from The European Food Safety Authority Nutrition Survey [55,56]. In a study conducted by Lutmer et al., for example, a variety of energy drinks were tested via GC, and some 88.9% (24 of 27) of which were found to contain low concentrations of ethanol [57]. Non-alcoholic foods should be studied not only because of their high sugar content [29,58] but also because they are sources of unintentionally consumed alcohol in the evaluation of clinical and forensic cases. However, in a Turkish study that considered the Turkish and German markets, the ethanol levels of non-alcoholic beverages in all samples were found to be below the allowed limit according to the corresponding codes [55,59]. Certain groups, such as children, pregnant women, and abstaining alcoholics, should be regarded as especially sensitive with respect to unintentional consumption and should thus be subjected to thoughtful evaluation. At the same time, the amount of ethanol widely used in herbal medicines was found to be several times higher than the lowest limit, suggesting that warnings are required for their administration to children [60].

In most cases, patients deny using alcohol when they have been suspected of alcohol abuse, which is a typical presentation of this disease [8]. Interestingly, even in free-choice alcohol selection situations, the levels of endogenous ethanol in rat blood and the alcohol preference of these animals are negatively correlated [13]. On the other hand, unwarranted alcohol consumption is known to increase the risk of developing liver cirrhosis and fatty liver disease [31]. If (conceivable) biases were excluded, some other cause of endogenous fermentation, such as gut microbiota dysbiosis, should be linked to liver disorders such as non-alcoholic fatty liver disease (NAFLD), non-alcoholic steatohepatitis (NASH), alcoholic liver diseases (ALDs), cirrhosis, and hepatic encephalopathy (HE). More thorough research singled out HiAlc KPN as an etiological factor of NAFLD in 60% of people [61]. ABS is observed among people with obesity-related liver disease, but apparently, healthy people are diagnosed as well [21,62]. Patients afflicted by other gastrointestinal conditions, such as gastroparesis, Crohn's disease, and short bowel syndrome, can also present with ABS [63,64]. Conversely, studies conducted on NAFLD patients suggest that there is a

bacterial origin of endogenous alcohol production, which might also be the causative micro-organisms in ABS cases.

### 2.4. Gut–Liver–Brain Axis and Forensic Alcohol Determination

The cause of ABS could be multifactorial; therefore, all possible causative factors must be carefully considered. There is a spotlight on the gut–liver–brain axis, which mediates the occurrence and development of many diseases and guides the research on disease treatment and, most certainly, forensic alcohol determination. The current understanding of the gut–liver–brain axis places the gut at the intersection of the brain and the liver, while this whole system is influenced by the gut microbiota [65,66].

Studies have reported that many patients experienced relief through dietary intervention and probiotics, while only a minority required antifungal therapy [11,34]. Accordingly, gut dysbiosis was first suggested by Eaton and Howard as the underlying cause of the gastrointestinal symptoms exhibited by some ABS patients [67]. An imbalance in the intestinal microbiota is associated with other gastrointestinal and systemic diseases. The combination of the enteric nervous system, the autonomic nervous system, and the central nervous system, with its neuroendocrine and neuroimmune features, is known as the microbiota–gut–brain axis. Due to the close anatomical and functional relations of the liver, the term microbiota–gut–liver–brain axis was introduced a few years back and has attracted increased attention, even with respect to the pathogenesis of ABS [8,10,50]. The most prominent component of this network is the mucus, which forms a protective physical barrier that prevents microorganisms and toxic substances from contacting the surface of the epithelium [68]. However, the disruption of this barrier may lead to inadequate colonization [49,69,70]. There is a missing piece of the puzzle with respect to the integrity of the gut–liver–brain axis. A protective physical barrier built up by mucus may succumb to mucus-eating microorganisms. Such microorganisms limit the interaction and penetration of bacteria, and a healthy mucus layer plays an important role in preventing diseases. Probiotics (a therapeutic approach administered in most ABS cases) modulate the properties of the mucus layer. This makes the gut microbiota a hot research topic with respect to its role as a possible ABS pathogenesis trigger (Table 2). Another microorganism from Table 2 found in gut microbiome strains is high-alcohol-producing Klebsiella pneumoniae (HiAlc KPN). This species has been strongly associated with the endogenous production of alcohol [69]. The distress of the gut microbiota challenges bowel habits. There is evidence that changing the gut microbiota through a fecal transplant often relieves symptoms such as diarrhea, indigestion, and abdominal pain. In this regard, it could be useful to evaluate whether isolated bacteria are mutated or the same as those found in most people.

Probably the most influential work on this topic was that of Hafez et al. They investigated BAC after carbohydrate ingestion in patients with diabetes mellitus (DM), liver cirrhosis (LC), and both DM and LC. In patients with LC but without DM, their BAC was significantly higher than that of the control but slightly lower than that in the DM (3.45 + 2.65 mg/dL) group. However, BAC and blood glucose levels were significantly correlated in each group (all groups) [21].

Studies in literature agree that the observed values of endogenously produced ethanol may not affect the brain function and ability of motor drivers [3,14,21,71].

### 2.5. Genetics

Humans worldwide report different experiences regarding human-produced alcohol, some of which have short- and long-term consequences, and many of these experiences are due to polymorphism in the genes whose product enzymes are responsible for alcohol metabolism. For instance, it has been shown that individuals with genetic polymorphisms of ADH and ALDH can find it arduous to metabolize ethanol, which can worsen alcohol intoxication symptoms [72]. Polymorphisms of the ADH and ALDH genes must be carefully considered in the legal setting. Specifically, it must be clear whether they could have contributed to the severity and development of ABS [73].

Even though it has not been studied with respect to ABS patients specifically, a genetic polymorphism that causes reduced aldehyde dehydrogenase enzyme activity has been identified [74–76]. For example, the presence of the homo-hypoactive genetic polymorphism in alcohol dehydrogen-ase1B (ADH1B) has been identified as being associated with Arg/Arg. Conversely, at the Glu487Lys of aldehyde dehydrogenase 2 (ALDH2) and in the corresponding polymorphism, homo-active Glu/Glu was identified [48]. This finding concerns a group of enzymes involved in the hepatic metabolism of ethanol. In relation to first-pass metabolism, this feature might explain the ethnic differences in the rates of endogenous ethanol production and clearance [23,33,77]. Variations in Fut-2, a gene encoding an enzyme responsible for the addition of terminal fucose residues to certain carbohydrates, change the gut carbohydrate environment. In the study conducted by Kashyap et al., it was reported that the specific Fut-2 gene mutation and a low-carbohydrate diet prevent the accumulation of the content in the intestine that might be a substrate for endogenous ethanol production [78].

In the context of the gut microbiota and ethanol production/carbohydrate fermentation, it is recognized that genetic susceptibility to Candida plays an important role in infections. Candida species members and other yeasts are normally natural defense mechanisms, and imbalances in the gut microbiota, or inadequate immunity, may lead to an increased susceptibility to invasive candidiasis [69]. Furthermore, a study testing stool genetically ("stool profiling") via polymerase chain reaction (PCR) at a stage when ethanol metabolites were found in urine revealed Saccharomyces cerevisiae (brewer's yeast) with a quintile distribution in the 3+ profile in addition to other micro-organisms [50], and these microorganisms are listed in Table 2 as possible causative microorganisms with respect to ABS.

### 2.6. Forensic Determination of Alcohol Concentration

Blood is the preferred specimen for determining alcohol concentrations, and the method for the detection of alcohol in blood was developed using head-space GC with flame ionization detection (HS-GC-FID) [9,79,80]. The results of toxicology assessments provide key information as to the type of substances present in an individual and the amount of these substances [81]. Having these data will enable experts to conclude whether these substances are consistent with a therapeutic dosage or are above a harmful level [82]. The most common equation used to estimate the blood alcohol concentration of an individual after their consumption of a known amount of alcohol is the Widmark equation.

The simplified version of the Widmark formula is as follows:

$$\text{BAC} = \left[ \frac{Consumed\ alcohol\,(\text{in grams})}{\text{Body weight in grams x r}} \right] \times 100 \tag{1}$$

In this formula, "r" is the gender constant, which equals 0.55 for females and 0.68 for males [12].

Time and again, the reliability of blood and breath tests is questioned, sometimes with the argument that alcohol can be produced naturally in the body [3,6,15]. However, this endogenous production, if in a spread-out form of a syndrome, significantly infringes on everyday life. Excessive colonization by fermenting microorganisms typically happens following disorders of the intestinal microbiome, such as in intestinal dysbiosis (most often of the fungal type). Recent antibiotic use is a possible cause of the change in gut microbiota preceding ABS [83]. However, the clinical manifestations can sometimes mimic food allergies or food intolerance [47]. The medical literature describes several strains of bacteria and fungi that may be associated with endogenous alcohol production [24,49]. Recently, an exceptional case of the endogenous production of alcohol in the oral cavity rather than in the intestines was presented [10,48].

However, the use of total body water is the preferred method in forensic blood alcohol calculations to assess whether the statutory limit of blood alcohol content has been exceeded rather than the ethanol's volume of distribution [84–86]. This is because alcohol has an

affinity for water, so the more water there is in which to distribute the alcohol, the lower the blood alcohol concentration [87]. Therefore, a person's blood alcohol concentration is a function of the total amount of alcohol in their system divided by total body water [85,86].

Postmortem Diffusion of Ethanol vs. Postmortem Microbiome Activity

In the framework of the current paper, the proper interpretation of postmortem ethanol analysis results should follow a step-by-step approach to estimate the suggested literature indicators, starting from those that are easiest to assess [88].

The estimation of postmortem blood ethanol depends on both antemortem and postmortem factors. The pharmacokinetics of the ingested beverage and the circumstances at the time of death, as well as factors such as the postmortem (via putrefaction) or in vitro production of ethanol after sampling, all influence blood ethanol estimation. Scientists remain discordant with respect to some aspects of postmortem blood ethanol estimation, specifically regarding which sampling site is the most appropriate or whether the water content of blood samples should be considered (mainly because of PM desiccation and the putrefactive process) [89,90]. Moreover, postmortem ethanol diffusion and/or redistribution can severely influence blood ethanol estimation [91,92].

Postmortem diffusion and the redistribution artifacts in relation to ethanol have long been described. Though the main mechanism might be diffusion along a concentration gradient, the mechanical properties of the organs also play a significant role [93].

It has been observed that postmortem ethanol is drawn most likely from solid organs (e.g., liver or lung) into the blood vessels and then into the cardiac chambers [94]. An ethanol shift from the stomach to the surrounding tissues has already been described [95,96]. This movement is typically described as "redistribution". This implies that the drug concentration in postmortem blood may not reflect the concentration before death [97]. On the other hand, "diffusion" is a chemical process resulting in redistribution.

Postmortem ethanol diffusion is sometimes subject to considerable site dependence. This mainly relates to the diffusion of ethanol from the stomach into the heart chambers between the time of death and sample collection, even with an intact gastric wall, with significant differences between heart and peripheral blood concentrations [88,98]. Postmortem diffusion could also be related to AM factors such as the time that one consumed their last drink before death, the quantity and strength of the beverage, and its dilution with food in the stomach. Estimating blood ethanol concentration based on the concentration from another sample (such as urine or vitreous humor) is quite unreliable.

Concerning the investigation of endogenous ethanol production following death, a lack of oxygen in the body results in cell autolysis, which releases macromolecules. The body's resident microbes, particularly those concentrated in the gastrointestinal tract, metabolize these cellular products in the process of putrefaction [99,100]. Thus, the detection of 1-propanol, isobutanol, methyl-butanols, and 1-butanol during a chromatographic ethanol analysis should imply the presence of microbially generated ethanol [88]. Even the living human gut microbiota produces large amounts of ethanol that might be clinically relevant [101].

All the other exact characteristics of a particular case, such as putrefaction state and clinical history, together with more discriminatory analyses providing more elaborate indicators of ethanol origin, such as ethyl-glucuronide, ethyl-sulfate, and serotonin metabolites, could be determined to provide complementary data that could help to identify an accurate interpretation of postmortem blood ethanol content [102].

### 2.7. Forensic Determination of Alcohol Concentration in Society 5.0

The extremely new concept of Society 5.0 is like a guide to social development and can have a profound impact on all points of society. It emphasizes the potential of the individual–technology relationship [103]. This leap forward furthers the improvement of the quality of life of all people in a sustainable world through a super-smart society [104]. Machine learning and artificial intelligence (AI) approaches are indispensable components

of this concept and have revolutionized multiple disciplines, including the forensic approach to BAC determination [105,106]. AI has the potential to outperform most forensic pathologists in toxicological laboratories [107]. Thus, it may soon automate and standardize the processes involved.

An AI-augmented forensic laboratory capable of BAC analysis offers the most elegant method of dodging spatial limitations. Space is needed in the forensic laboratory as there is a need to store several thousands of images and physical archives used for other forensic analyses [108,109]. On the other hand, the plots used in AI datasets are visually different from the original; however, the reasonable demands for data storage make artificial neural networks (ANNs) optimal in this regard [9,15,16,110].

For this purpose, as well as in forensic science analysis for any other purpose, AI-influenced ANNs can be employed. Furthermore, in the context of exploring all metabolites (or products of a "bioreactor") for a digitalized and fully developed society without time-wasters, approaches like "metabolomics" or "multi-omics" should be pursued [111].

The present state-of-the-art toxicological techniques are far from relying on some anecdotal attainments [112–114]. Learning from the input data mimics the functioning of neurons and their communications to convey complex behavior. This progress follows the progress in the preparation and processing of input material, where the accessibility of large datasets goes hand-in-hand with the expansion in algorithm structures. On the other hand, progress in computing programming has ignited diligence with respect to learning the AI-constructed machines built for the high-dimensional output of data [115]. A model for the near future consists of a forensic medicine specialist skilled in toxicology enhanced with a real-time artificial intelligence system's second review [116,117].

Thus, a novel strategy of deep learning incorporated in toxicological laboratories should be proposed. Machines should be trained and validated with respect to toxicology in subjects that cover diverse and representative clinical cases, as commonly seen in everyday practice [118]. This would result in an AI system that can handle large numbers of toxicological reports without the potential disturbances commonly experienced by professionals in the field (e.g., space or time limitations) [119,120]. Such a system would drastically alleviate the heavy clinical burden of daily work and would also be a generalizable tool for other professions with similar background knowledge.

Deep learning AI models are currently used in analytical procedures as an assessment tool to help with efficiency, consistency, and decision making. Unfortunately, a forensic specialist skilled in toxicology still needs to be at the center of such an assessment. By all accounts, this will remain the case, at least in the near future [121,122].

### 2.8. Comprehensive Evaluation

Clinical conditions are highly recommended for the diagnostics of gut fermentation to minimize the possible negative effects of this syndrome in the event of its occurrence in full [24,50,123]. Even though clearly defined and standardized criteria for establishing an unambiguous diagnosis of ABS do not exist, based on the available medical literature, for the purpose of this review, a list of procedures was developed (Figure 3) [34].

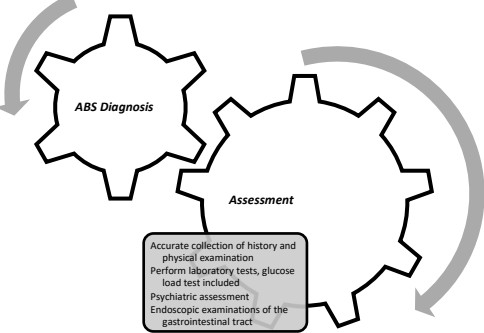

**Figure 3.** Diagram showing rational assessment of suspected ABS in a patient.

All relevant data related to a patient's history and physical examinations must be collected most attentively and in a detailed manner. Possible concomitant diseases should be carefully considered, and their possible connection to episodes of ABS should be well explained. A patient's typical diet and consumption of alcoholic beverages must be considered [124]. Psychiatric assessments should also be mandatory in order to eliminate psychiatric diseases and the so-called 'secret drinking' of alcoholic beverages [125,126]. Laboratory tests should also be performed with great scrutiny. Microbiological tests of tissues and feces should be carefully assessed [24,49,50,52], and urine and the oral cavity must be most carefully evaluated [10,48,51,127]. There are several alcohol-producing species of microorganism whose colonization might help to identify the causal organism of ABS. The identification of the microorganisms listed in Table 2 could direct clinicians toward adequate therapy [8,128].

**Table 2. Possible causative microorganisms** [8,11,20,24,25,51,61,63,64,91].

| Genera of Causative Microorganisms |
| :---: |
| *Saccharomyces* spp. |
| *Candida* spp. |
| *Klebsiella* spp. (*pneumoniae*) |
| *Escherichia* spp. |
| *Streptococcus* spp. |
| *Bacteroides* spp. |
| *Bifidobacterium* spp. |
| *Clostridium* spp. |
| *Pseudomonas* spp. |

The so-called glucose load test should be conducted at certain time intervals. In this test, an organism is provoked to produce ethyl alcohol in the blood [127,129]. Some authors recommend a challenge of 100–200 g of glucose combined with blood alcohol concentration (BAC) and breath or plasma alcohol testing at intervals of 0, 4, 8, 16, and 24 h [8]. The results of this test should not be declared negative until it is completed thoroughly because the delayed testing of samples after 16 and 24 h might be crucial in cases where the conversion process by fungi takes up to 24 h or longer, as is the case with some fungi [10]. Scientists' opinions are not in unison with respect to this test, as there are disagreements as to whether the results of this test can be declared negative unless it is performed as outlined above. However, in one study, 40 g of glucose was administered to a patient, and their blood alcohol levels were measured hourly for 6 h. Based on this, the production of alcohol from the intestinal tract was ruled out as a cause of his seizures [48,127].

Finally, it is rational to perform endoscopic examinations of the upper (e.g., gastroscopy) and lower (e.g., colonoscopy) gastrointestinal tract and collect microbiological testing material. However, there are cases where the microorganisms listed in Table 2 have been identified in gastric and jejunal samples. *K. pneumoniae* was found in gastric and jejunal samples in the case submitted by Saverimuttu et al. [50]. However, in their case, *K. pneumoniae* was persistently present and part of the mixed flora.

### 2.9. Treatment Options

Generally, a low-carbohydrate diet has been a sufficient treatment for ABS. In a few case reports, fluconazole was administered for 3 weeks. In some other cases, itraconazole, voriconazole, metronidazole, or combinations were used. A number of case reports described recent antibiotic use before or at the onset of symptoms [8]. Therefore, it is quite possible that the use of antibiotics might affect the microflora and allow for the colonization of alcohol-producing species [34].

*2.10. Fecal Microbiota Transplantation*

Recent antibiotic use is reported to be a possible cause of the change in gut microbiota that is thought to precede ABS. Likewise, any other abdominal conditions such as surgery or strictures—which are thought to alter abdominal anatomy, thus facilitating fungal or bacterial overgrowth—might be blamed for endogenous ethanol production [70,83].

Fecal microbiota transplantation (FMT) is the relocation of stool (via transferal) from a healthy donor into the colon of a patient with altered microbiota [122]. The process involves the restoration of the colonic microflora by introducing healthy bacterial flora, which, in some cases, is freeze-dried [130,131].

There are few cases of the successful treatment of ABS in the relevant literature. The first successful treatment of a patient with chronic gut fermentation syndrome via FMT was described in Belgium in 2020 [70]. The described case appears to have been successful, even after all other therapies had failed. However, experts await further studies on this treatment modality. This is rational, however, as ABS is not an "on-label" indication for FMT use. Physicians are often alarmed by the reason behind whether they can defend themselves from possible legal charges claiming that they deviated from the standard of care without need [3,5,132,133].

As in the case of any weakly studied therapeutic approach, even with FMT, it is difficult to balance theoretical long-term harms against direct benefits. This is a rare case where patients can try certain remedies with no clinical supervision.

## 3. Conclusions

Forensic experts and law professionals should remain broad-minded regarding the possibility of endogenous ethanol production; however, they should also not overestimate ABS. Bear in mind that it is a rare and often misunderstood and unrecognized condition. Fermentation in an organism acts as a "human bioreactor" where ingested carbohydrates are converted into alcohol. The concentrations of endogenously produced ethanol are far too low to have any legal or medical significance.

However, because of the legal implications and the lack of studies on ABS offering high-level evidence, every other possibility must be eliminated prior to diagnosing ABS. A diagnosis can be made through adequate history taking and a carbohydrate challenge test (glucose load test). The reporting of blood ethanol levels measured in secure hospital environments, where alcoholic beverages would not be obtainable, would constitute useful research. The existing literature mainly consists of case reports and bears the weight of a high risk of potential bias.

Various co-morbidities underlying ABS, such as gastroparesis, Crohn's disease, short bowel syndrome, and obesity-related liver disease, have been reported and so must be considered. Moreover, our understanding of the factors that cause or even contribute to ABS is still blurred. Even so, the *Saccharomyces* and *Candida* genera have been recognized as the culprits behind this condition. In some cases, previous treatment with antibiotics has been conducted.

The primary treatment for ABS is to follow an appropriate diet until symptoms subside. The use of probiotics is also recommended, and most cases of this syndrome, especially low-grade ("subtle") cases, do not require other therapies, although current treatments also include antifungal medications. The literature implies that there is a potential role of FMT in the treatment of this syndrome after all other therapies have failed.

**Funding:** This research received no external funding.

**Institutional Review Board Statement:** Not applicable.

**Informed Consent Statement:** Not applicable.

**Data Availability Statement:** Available on request.

**Acknowledgments:** This author acknowledges the University of Rijeka, Faculty of Medicine, for their constant support.

**Conflicts of Interest:** The funder had no role in the design of the study; in the collection, analyses, or interpretation of data; in the writing of the manuscript; or in the decision to publish the result.

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
