# Peer review of "The Human Body as an Ethanol-Producing Bioreactor—The Forensic Impacts"

_fermentation, doi:10.3390/fermentation9080738_

Round 1

Reviewer 1 Report

The title is somehow misleading and should be modified. Also, the authors should focus on the main target of this review, "endogenous ethanol production and their respective forensic impacts."

Lines 36-37: Provide a reference.

Table 1: What is the source of these data?

Line 51: Show the metabolism of alcohol as a diagram

Lines 62-63: Dietary recommendations for who? Is it suitable for all ages?

Line 64: it is not clear.

Lines 76-78: What is the period covered by the research? Or not specified?

Lines 95-103: check the format of this diagram. Some words are not shown.

Line 128: This section should be detailed with more experimental conditions for each presented result. Also, the concentrations of endogenously produced ethanol should be clarified.

Figure 2 is not clear.

Table 2: Reference.

Line 511: Write scientific names in italic.

Line 546: The authors should refer to the the rates of ABS cases.

Minor editing of English language required

Author Response

The author acknowledges the reviewer's time and efforts. The title of this paper has been changed as suggested and the changes in the maintext have also been made accordingly. The section on the "Society 5.0" has been totally removed, as it is ambiguously written and confusing in the present format. 

Reviewer 2 Report

The writing of the paper was good with no issues around English and grammar. The work was informative highlighting the issue of Auto-brewery syndrome. it does show how the body operates like a brewery/bioreactor producing ethanol from sugar present in food eaten and how the bacteria convert this to ethanol.

In figure 1 some of the words are cut off. 

The bacteria involved are they found in unusual abundance?

are they mutated or are they the same as in everyone else?

Do they produce in the stomach? because it is acidic hence the pH is low, so are they adjusted to this. 

Author Response

The writing of the paper was good with no issues around English and grammar. The work was informative highlighting the issue of Auto-brewery syndrome. it does show how the body operates like a brewery/bioreactor producing ethanol from sugar present in food eaten and how the bacteria convert this to ethanol.

1.In figure 1 some of the words are cut off.

Thank you for this kind compliment.

2.The bacteria involved are they found in unusual abundance?

Thank you for noticing, this was corrected.

3.are they mutated or are they the same as in everyone else?

Thank you for asking, the exact number of colonies was not given in the works used.

4.Do they produce in the stomach? because it is acidic hence the pH is low, so are they adjusted to this.

Exact genotypes were not provided. However, this sounds like awesome idea for further research, so, if it is ok with this reviewer this idea will be included in the MS.

Thank you for this idea. In the present literature there is no such information.

Reviewer 3 Report

This review of ABS is valuable work, however, it needs some further work:

There are some formatting issues:

· The paragraph "conclusion" is too narrow. It needs to be adjusted

· Table 1 is not in the required format of MDPI (e.g., there are vertical lines)

· Figure 1 needs to be corrected (it has to be aligned to the text, some of the text is not visible because it is larger than the box, and the blue boxes are out of the line)

It would be good to include a table describing how high BAC was achieved in the reported cases of ABS found during the review of the literature with some other information about the given patient (age, sex, suspected cause of ABS)

The author mentions the legal/forensic relevance of this topic (even in the abstract), but misses the forensic evaluation – how a forensic specialist can determine what amount of BAC could come from ABS in a given case. Are there any toxicological findings which help the distinction (other by-products of fermentation in case of ABS detectable by GC-MS-FID – as it is available if the question is the port-mortem ethanol production)? That would be the most important value of the manuscript. Although clinical tests are described in the subsection „Comprehensive evaluation”, these test are very limited in a Forensic situation.

I do not see the significance of the subsection „Forensic Determination of Alcohol Concentration in Society 5.0” in this form. It does not connect closely to the topic. It should be deleted or expanded to explain its significance in the given topic (e.g. how it can help to determine whether the BAC is from alcohol consumption or comes from ABS).

The manuscript is easy to read, but there are some mistakes (colon-semicolon, wrong preposition, typo), and inconsistencies have to be corrected (like inconsistent hyphenation e.g. by-product and byproduct) and also errors in the reference list (e.g. lines 762, 794, 797, 828). A thorough check is necessary.

Author Response

(The authors gave the same response as above.)
